# Duration-Dependent Risk of Hypoxemia in Colonoscopy Procedures

**DOI:** 10.3390/jcm13133680

**Published:** 2024-06-24

**Authors:** Eyal Klang, Kassem Sharif, Offir Ukashi, Nisim Rahman, Adi Lahat

**Affiliations:** 1Division of Data-Driven and Digital Medicine (D3M), Icahn School of Medicine at Mount Sinai, New York, NY 10029, USA; eyalkla@hotmail.com; 2Department of Gastroenterology, Sheba Medical Center, Affiliated with Tel Aviv University Medical School, Tel Hashomer, Ramat Gan 52621, Israel; kassemsharif@gmail.com (K.S.); offirukashi@gmail.com (O.U.); 3ARC Innovation Center, Sheba Medical Center, Affiliated with Tel Aviv University Medical School, Tel Hashomer, Ramat Gan 52621, Israel; nisim.rahman@sheba.health.gov.il; 4Department of Gastroenterology, Samson Assuta Ashdod Medical Center, Affiliated with Faculty of Medicine, Ben Gurion University of the Negev, Be’er Sheva 84105, Israel

**Keywords:** hypoxemia, colonoscopy, sedation, duration

## Abstract

**Background and Aims**: Colonoscopy is a critical diagnostic and therapeutic procedure in gastroenterology. However, it carries risks, including hypoxemia, which can impact patient safety. Understanding the factors that contribute to the incidence of severe hypoxemia, specifically the role of procedure duration, is essential for improving patient outcomes. This study aims to elucidate the relationship between the length of colonoscopy procedures and the occurrence of severe hypoxemia. **Methods**: We conducted a retrospective cohort study at Sheba Medical Center, Israel, including 21,524 adult patients who underwent colonoscopy from January 2020 to January 2024. The study focused on the incidence of severe hypoxemia, defined as a drop in oxygen saturation below 90%. Sedation protocols, involving a combination of Fentanyl, Midazolam, and Propofol were personalized based on the endoscopist’s discretion. Data were collected from electronic health records, covering patient demographics, clinical scores, sedation and procedure details, and outcomes. Statistical analyses, including logistic regression, were used to examine the association between procedure duration and hypoxemia, adjusting for various patient and procedural factors. **Results**: We initially collected records of 26,569 patients who underwent colonoscopy, excluding 5045 due to incomplete data, resulting in a final cohort of 21,524 patients. Procedures under 20 min comprised 48.9% of the total, while those lasting 20–40 min made up 50.7%. Only 8.5% lasted 40–60 min, and 2.9% exceeded 60 min. Longer procedures correlated with higher hypoxemia risk: 17.3% for <20 min, 24.2% for 20–40 min, 32.4% for 40–60 min, and 36.1% for ≥60 min. Patients aged 60–80 and ≥80 had increased hypoxemia odds (aOR 1.1, 95% CI 1.0–1.2 and aOR 1.2, 95% CI 1.0–1.4, respectively). Procedure durations of 20–40 min, 40–60 min, and over 60 min had aORs of 1.5 (95% CI 1.4–1.6), 2.1 (95% CI 1.9–2.4), and 2.4 (95% CI 2.0–3.0), respectively. **Conclusions**: The duration of colonoscopy procedures significantly impacts the risk of severe hypoxemia, with longer durations associated with higher risks. This study underscores the importance of optimizing procedural efficiency and tailoring sedation protocols to individual patient risk profiles to enhance the safety of colonoscopy. Further research is needed to develop strategies that minimize procedure duration without compromising the quality of care, thereby reducing the risk of hypoxemia and improving patient safety.

## 1. Introduction

In the United States, sedation is administered in more than 98% of colonoscopy procedures to mitigate patient anxiety and discomfort, increase diagnostic accuracy, and minimize memory of the procedure [1]. The choice and amount of sedation are precisely tailored to suit the individual needs of each patient and the specifics of the colonoscopy, aiming to maximize safety, patient comfort, and procedural effectiveness.

During moderate sedation, patients are capable of maintaining spontaneous respiration and cardiovascular stability, with the ability to react to verbal commands or physical prompts [1,2]. Deep sedation, however, significantly reduces a patient’s responsiveness and may require interventions to ensure the airway remains open.

Hypoxemia, characterized by a drop in arterial oxygen levels below the normal range (PaO2 < 80 mm Hg, saturation < 95%), is a risk associated with these procedures. Severe hypoxemia is defined in the literature as a measurement of PaO2 < 60 mm Hg and saturation < 90% [3]. It can lead to critical complications, such as arrhythmias, neurological damage, myocardial ischemia, respiratory failure, or even death.

The incidence of hypoxemia varies greatly (from 4% to 70%), influenced by factors including patient diversity, endoscopic practices, and anesthetic choices [3,4,5]. Specifically, in colonoscopies, factors such as patient positioning and the physiological effects of sedatives can affect respiratory function, potentially leading to decreased oxygen saturation [5,6].

Effective strategies to mitigate this risk are critical for improving patient safety. A recent study suggested that high-flow nasal oxygen (HFNO) might offer superior outcomes in maintaining adequate oxygen levels during endoscopic procedures in high-risk patients [7].

We hypothesize that the duration of a medical procedure, particularly one as intricate as a colonoscopy, may significantly influence the risk of developing hypoxemia. Extended procedural times are associated with prolonged periods of sedation, which in turn can exacerbate the risk of respiratory depression, a primary contributor to hypoxemia.

Therefore, in our current study, we aim to examine the influence of colonoscopy duration on the occurrence of hypoxemia in a large tertiary center cohort.

## 2. Materials and Methods

### 2.1. Study Design and Population

This retrospective cohort study was conducted at Sheba Medical Center, a large tertiary center in Israel. The study included all adult patients who underwent colonoscopies between January 2020 and January 2024. We excluded patients with incomplete data.

The primary endpoint was the incidence of severe hypoxemia [3], defined as a drop in oxygen saturation level below 90% during the procedure.

This study was conducted and reported in accordance with the Strengthening the Reporting of Observational Studies in Epidemiology (STROBE) guidelines.

### 2.2. Colonoscopy Procedure

At our institution, colonoscopic procedures are performed using Olympus 190 series by both experts in gastroenterology and trainees under experts’ supervision. Colon is insufflated by CO_2_.

Sedation is administered by the endoscopist and usually includes a combination of sedatives comprising fentanyl, midazolam and propofol. Sedation is given gradually, in order to minimize discomfort and anxiety and maintain patients’ tolerance. The choice of sedation protocol is determined by the endoscopist’s discretion, showcasing personalized approaches to patient care management. All patients are treated prophylactically with O_2_ 3 L/min given in nasal cannulas and are monitored closely through the examination for vital signs (blood pressure, pulse, and saturation).

All measurements are registered in the electronic file. saturation is measured continuously, and registered electronically every 5 s. Endoscopy duration is measured and registered. Hypoxemia events are treated per the protocol and according to clinical guidelines and escalate from an elevation of O_2_ flow and chin lift to O_2_ delivery through reservoir mask or full resuscitation.

### 2.3. Data Collection

Data were collected using standard Structured Query Language (SQL) procedures from our hospital’s electronic database. Data included patient demographics, clinical assessment scores (an American Standards Association (ASA) score and Mallampati score), details of anesthetic medication administration and dosage (propofol, midazolam, fentanyl), endoscopy duration, records of failed colonoscopies, records of biopsies during the procedure, and intra-procedural oxygen saturation levels.

### 2.4. Ethical Considerations

The study received approval from the Institutional Review Board (IRB) of Sheba Medical Center. In line with the retrospective design of the study, informed consent was waived.

### 2.5. Statistical Analysis

Categorical variables were analyzed using percentages and the Chi-square test. Continuous variables were assessed by reporting both the median with interquartile range (IQR) and mean with standard deviation (SD) and evaluated using the one-way ANOVA test. A *p*-value of less than 0.05 was considered statistically significant.

We employed logistic regression analysis to elucidate the relationship between procedure duration and hypoxemia incidence, adjusting for confounders including age, gender, procedure type, ASA classification, Mallampati score, and medication dosages. Procedure durations were stratified into quartiles: <20, 20–40, 40–60, and >60 min. Results are presented as adjusted odds ratios (AORs) with 95% confidence intervals (CIs) and associated *p*-values, facilitating a nuanced interpretation of risk factors.

All statistical analyses were performed using Python version 3.10.

## 3. Results

We initially collected records of 26,569 patients who underwent colonoscopies at our institution. We removed 5045 patients due to incomplete data. Our final cohort included 21,524 patients (81.7% of the original number).

The graph in Figure 1 illustrates the frequency of colonoscopies across different duration categories. The majority of procedures fall within the shorter durations, with 9259 colonoscopies (48.9%) completed in less than 20 min and 9591 (50.7%) within 20 to 40 min. There is a notable decrease in frequency as the duration increases, with 1606 procedures (8.5%) lasting between 40 and 60 min and only 543 (2.9%) extending beyond 60 min.

We observed differences in patient characteristics and outcomes based on the colonoscopy duration (Table 1).

In our study, longer colonoscopies were linked to a higher risk of low oxygen levels. For short procedures (under 20 min), 17.3% had hypoxemia. This increased to 36.1% for procedures over 60 min. Patients were older in longer procedures, with median ages from 56 to 67 years. More medicine, like propofol, was used in longer endoscopies, up to 380.3 mg on average for the longest. Notably, during longer procedures, more biopsies were obtained, reaching 80.5% in 40–60 min procedures.

The logistic regression analysis demonstrated significant predictors of hypoxemia during colonoscopy. Age and colonoscopy duration were notable factors. Patients aged 60–80 and ≥80 had increased odds of hypoxemia, with adjusted odds ratios (aOR) of 1.1 (95% CI 1.0–1.2) and 1.2 (95% CI 1.0–1.4), respectively. Duration of the procedure emerged as a strong predictor: durations of 20–40 min, 40–60 min, and over 60 min were associated with progressively higher odds of hypoxemia, with aORs of 1.5 (95% CI 1.4–1.6), 2.1 (95% CI 1.9–2.4), and 2.4 (95% CI 2.0–3.0), respectively. Other variables, such as sex, elective procedure status, and medication dosages (midazolam, propofol, fentanyl), had less pronounced effects. The forest plot in Figure 2 visually represents the aORs and their respective confidence intervals, reinforcing the relationship between increased colonoscopy duration and the risk of severe hypoxemia.

Figure 3 displays the percentage of severe hypoxemia events across different colonoscopy durations stratified by age groups. For the youngest cohort (<40), hypoxemia rates rise from 18.3% for procedures under 20 min to 43.2% for those over 60 min. This increasing trend persists across all age categories, with the 40–60 age group experiencing hypoxemia rates from 16.7% to 33.1%, the 60–80 age group from 19.8% to 35.3%, and the oldest group (≥80) from 22.5% to a significant 40.0% for the longest durations. These percentages emphasize the heightened risk of severe hypoxemia with longer colonoscopy procedures across all age groups.

## 4. Discussion

In this study, performed on a large cohort of more than 20,000 patients, we elucidated the significant relationship between the duration of colonoscopy procedures and the incidence of hypoxemia, contributing to the evolving discourse on procedural safety and patient outcomes in gastroenterological practices.

Our findings reveal a clear trend: as colonoscopy duration extends, the risk of hypoxemia markedly increases. Specifically, short procedures (under 20 min) exhibited a 17.3% incidence of hypoxemia, which escalated to 36.1% for procedures exceeding 60 min. This gradient of risk underscores the procedural duration as a critical factor in patient safety protocols.

The demographic and clinical characteristics of our patient cohort, particularly age and the volume of sedatives administered, notably influenced the risk profile for hypoxemia. Older patients, especially those aged 60–80 and ≥80, demonstrated progressively higher odds of experiencing hypoxemia, with adjusted odds ratios (aOR) of 1.1 and 1.2, respectively. This age-related vulnerability to hypoxemia is consistent with the physiological changes associated with aging, such as reduced pulmonary reserve and altered pharmacodynamics, which can exacerbate the effects of sedatives like propofol [7].

Our results are in line with current literature, showing that hypoxemia during colonoscopies may develop in 26–85% of procedures, arising from a mix of factors including sedation-caused collapse of the upper airway, respiratory depression, and the compression of the lungs due to the introduction of gas into the intestines [8,9,10,11].

Patients suffering from obstructive sleep apnea syndrome, obesity, hypertension, diabetes, heart disease, those aged over 60 years, or those categorized within a higher American Society of Anesthesiologists (ASA) physical status class are especially vulnerable to hypoxemia and its associated risks [8,9,10,11].

Indeed, our analysis showed a significant increase in propofol dosage in longer procedures, reaching an average of 380.3 mg for the longest durations, which likely contributed to the observed increase in hypoxemia risk.

Moreover, the logistic regression analysis highlighted colonoscopy duration as a potent predictor of hypoxemia, with durations of 20–40 min, 40–60 min, and over 60 min associated with adjusted Odd Ratios of 1.5, 2.1, and 2.4, respectively.

This relationship between procedure length and hypoxemia incidence is a critical finding, suggesting that procedural efficiency and the judicious use of sedation could play significant roles in mitigating this risk.

Interestingly, the increase in biopsies obtained during longer procedures, which peaked at 80.5% for durations between 40 and 60 min, suggests a procedural complexity that may necessitate extended durations. This complexity, while necessary for thorough diagnostic evaluation, underscores the need to balance procedural thoroughness with patient safety considerations.

Our study’s stratification of severe hypoxemia events by age group further illuminates the pervasive risk across all demographics, with notable increases in incidence correlating with both age and procedure length. Such findings advocate for heightened vigilance and tailored sedation protocols, especially for older patients and those undergoing longer procedures.

These findings align with the findings of Kim et al. [12], who demonstrated that pre-oxygenation before endoscopic sedation significantly reduces the incidence of hypoxia in elderly patients.

Our data similarly underscore the multifactorial nature of hypoxemia during sedation endoscopy, reinforcing the importance of considering patient-specific factors such as age, comorbidities, and baseline respiratory function.

Interestingly, our analysis did not find a significant direct correlation between the specific sedation protocols used (Fentanyl, Midazolam, and Propofol) and the incidence of severe hypoxemia. This observation might be explained by the fact that the administration of sedatives was personalized based on the endoscopist’s discretion, allowing for variability in dosing and combination of drugs tailored to individual patient needs and responses. This personalization means that any potential effects of the sedative drugs on hypoxemia could be mitigated by careful monitoring and adjustment by experienced clinicians. These results are consistent with a recent meta-analysis, which found that Propofol sedation poses a similar risk of cardiopulmonary adverse events compared to traditional agents used in gastrointestinal endoscopic procedures [13].

While our study provides valuable insights into the relationship between colonoscopy duration and the incidence of hypoxemia, several limitations warrant consideration. Firstly, the retrospective nature of our study inherently limits our ability to establish causality between procedure duration and hypoxemia risk. Although we adjusted for a range of confounders in our analysis, the potential for unmeasured variables to influence outcomes cannot be fully excluded. For instance, detailed information on intra-procedural patient movements and posture or the precise timing of sedative administration throughout the procedure was not available. Furthermore, our database encompassed procedures performed by over 30 different endoscopists over the years. Although it was not structured to account for variations in their experience and training levels, this diversity covers a wide range of expertise, enhancing the generalizability and robustness of our results by reflecting real-world practice conditions.

Secondly, our reliance on electronic health records for data extraction may introduce bias related to the accuracy and completeness of recorded information. Furthermore, our study population, derived from a single center, may limit the generalizability of our findings to other settings or populations with different demographic or clinical characteristics. Finally, while we observed a significant association between longer procedure times and increased hypoxemia risk, the clinical significance of these findings requires further exploration to understand the impact on patient outcomes and to develop targeted interventions. Despite these limitations, our study contributes to the growing body of evidence on procedural risk factors for hypoxemia during colonoscopy and underscores the need for continued research in this area.

In conclusion, our study highlights the imperative for gastroenterologists to consider the duration of colonoscopy procedures as a significant factor in the risk management of hypoxemia. The implications of our findings suggest that efforts to optimize procedural efficiency, customize sedation practices to patient-specific risk factors, and employ continuous monitoring strategies are paramount in enhancing the safety of colonoscopy. Future research should focus on developing and validating interventions that can reduce the duration of procedures without compromising diagnostic and therapeutic outcomes, thereby minimizing the risk of hypoxemia and improving patient care.

## Figures and Tables

**Figure 1 jcm-13-03680-f001:**
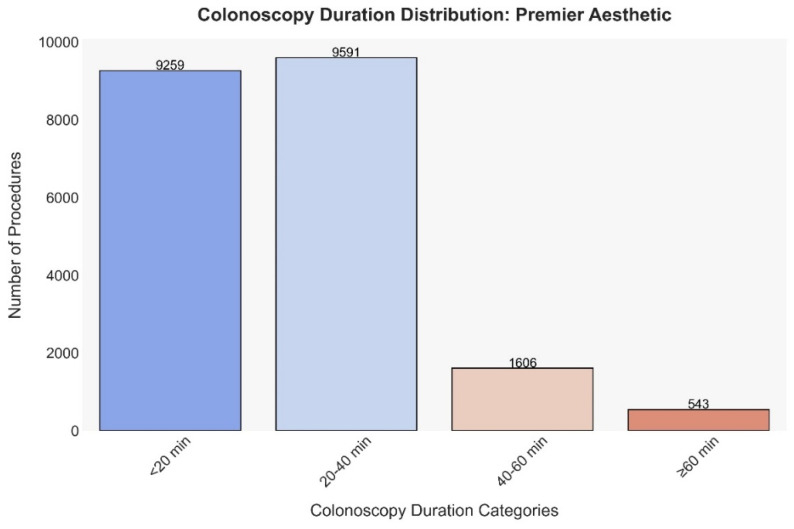
Frequency of colonoscopies across different duration categories.

**Figure 2 jcm-13-03680-f002:**
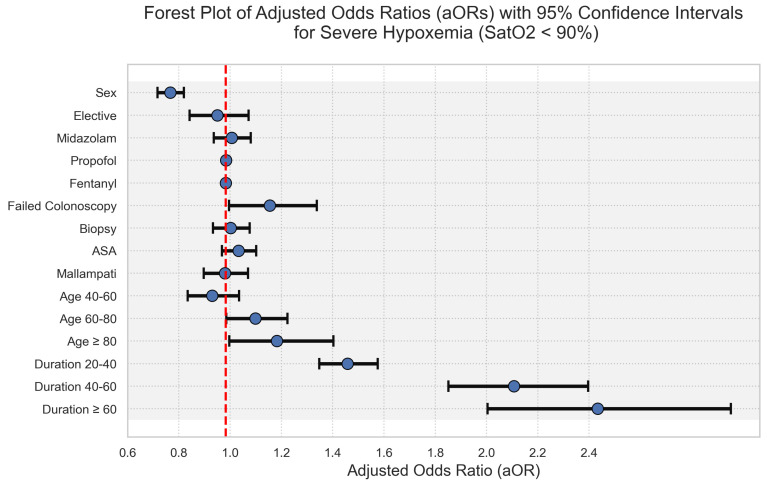
Forest Plot of Adjusted Odds Ratios (aOR) with confidence intervals for Hypoxemia Risk Factors. This plot shows the aORs for sex, inpatient status, ASA score, Malampati score, medication dosages, failed colonoscopy, biopsy, age groups, and duration of the procedure. Age group < 40 y and duration < 20 min serve as baseline and thus are not presented on the graph (both aOR 1.0). Points indicate aORs; horizontal lines represent 95% confidence intervals.

**Figure 3 jcm-13-03680-f003:**
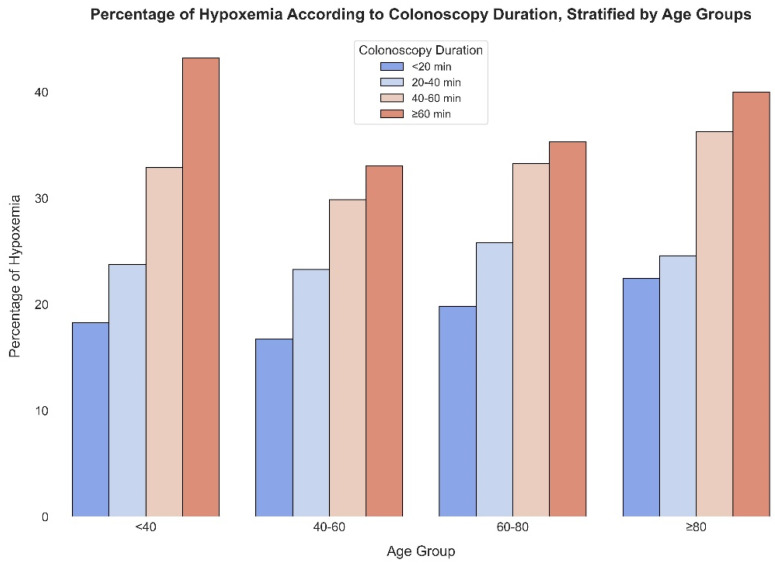
Hypoxemia rates (SatO2 < 90%) in different age groups, comparing different procedure duration.

**Table 1 jcm-13-03680-t001:** Patient characteristics and outcomes based on colonoscopy duration.

	Duration < 20 min	Duration 20–40 min	Duration 40–60 min	Duration ≥ 60 min	*p*-Value
Age (years)	Median 56.0, IQR (42.4–68.1), Mean 54.1, SD ± 17.8	Median 62.7, IQR (50.9–71.9), Mean 59.8, SD ± 16.0	Median 68.6, IQR (59.4–74.8), Mean 66.1, SD ± 12.9	Median 67.0, IQR (56.5–74.9), Mean 64.4, SD ± 14.9	<0.001
Sex (n, %)	4814, (54.1%)	5446, (52.8%)	1001, (57.8%)	343, (59.2%)	<0.001
Inpatient (n, %)	797, (9.0%)	832, (8.1%)	177, (10.2%)	61, (10.5%)	<0.001
ASA Score	Median 2.0, IQR (2.0–2.0), Mean 1.9, SD ± 0.6	Median 2.0, IQR (2.0–2.0), Mean 2.0, SD ± 0.5	Median 2.0, IQR (2.0–2.0), Mean 2.1, SD ± 0.5	Median 2.0, IQR (2.0–2.0), Mean 2.1, SD ± 0.6	<0.001
Mallampati Score	Median 2.0, IQR (2.0–2.0), Mean 1.8, SD ± 0.4	Median 2.0, IQR (2.0–2.0), Mean 1.8, SD ± 0.4	Median 2.0, IQR (2.0–2.0), Mean 1.8, SD ± 0.4	Median 2.0, IQR (2.0–2.0), Mean 1.9, SD ± 0.4	<0.001
Midazolam (mg)	Median 2.0, IQR (2.0–2.0), Mean 1.9, SD ± 0.6	Median 2.0, IQR (2.0–2.0), Mean 1.9, SD ± 0.5	Median 2.0, IQR (2.0–2.0), Mean 1.9, SD ± 0.5	Median 2.0, IQR (2.0–2.0), Mean 1.9, SD ± 0.6	<0.001
Propofol (mg)	Median 200.0, IQR (100.0–200.0), Mean 185.9, SD ± 95.8	Median 200.0, IQR (160.0–300.0), Mean 229.9, SD ± 118.2	Median 300.0, IQR (200.0–400.0), Mean 300.6, SD ± 163.8	Median 300.0, IQR (200.0–500.0), Mean 380.3, SD ± 259.1	<0.001
Fentanyl (mcg)	Median 50.0, IQR (0.0–50.0), Mean 38.9, SD ± 28.2	Median 50.0, IQR (50.0–50.0), Mean 44.0, SD ± 26.4	Median 50.0, IQR (50.0–50.0), Mean 47.4, SD ± 30.0	Median 50.0, IQR (50.0–50.0), Mean 45.8, SD ± 30.2	<0.001
Failed Colonoscopy (n, %)	897, (10.1%)	284, (2.8%)	32, (1.8%)	12, (2.1%)	<0.001
Biopsy (n, %)	3663, (41.1%)	6937, (67.3%)	1395, (80.5%)	437, (75.5%)	<0.001
SatO2 < 90%	1540, (17.3%)	2494, (24.2%)	562, (32.4%)	209, (36.1%)	<0.001

## Data Availability

The authors declare that there is no relevant data available for this study. All data used in the analysis and preparation of this manuscript have been included in the manuscript.

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
