# Peer review of "Duration-Dependent Risk of Hypoxemia in Colonoscopy Procedures"

_jcm, 2024, doi:10.3390/jcm13133680_

Round 1
Reviewer 1 Report
Comments and Suggestions for Authors
This is a retrospective study with a large cohort of patients evaluating the incidence of severe hypoxemia in patients undergoing colonoscopy. So far, results do not provide further informations on this topic, I think that the paper could be enriched with the analysis of risk factors associated with hypoxemia, it could be interesting evaluate the risk on the basis of patients clinical characteristics and sedation type.
Author Response
Dear reviewer, we thank you for your insightful comments, and for the opportunity to revise the manuscript . We feel the manuscript is now much improved.
We present here a point-by-point response to your comments. For each comment, we copy the original text and include our responses. When changes have been made to the manuscript, we make sure to cite them here, so as to facilitate review of the changes.
Your review:
This is a retrospective study with a large cohort of patients evaluating the incidence of severe hypoxemia in patients undergoing colonoscopy. So far, results do not provide further informations on this topic, I think that the paper could be enriched with the analysis of risk factors associated with hypoxemia, it could be interesting evaluate the risk on the basis of patients clinical characteristics and sedation type.
Our response:
Dear reviewer, we thank you for your comment.
Notably, both ASA score and Mallampati score were assessed as indicators of patients' clinical characteristics, along with the types and dosages of sedation medications. This was detailed in the methods section (page 3, lines 100-103, highlighted text) and evaluated in the results section, as shown in Table 1 and Figure 2 (forest plot).
The relevant text in the manuscript:
Data included patient demographics, clinical assessment scores (A American Standards Association (ASA) score and Mallampati score), details of anesthetic medication administration and dosage (propofol, midazolam, fentanyl).
See also Table 1 and Figure 2.
Reviewer 2 Report
Comments and Suggestions for Authors
Dear Authors,
The manuscript deals with a very interesting and clinically relevant topic.
The manuscript is written in a clear and concise manner.
I think it would be interesting to investigate the impact of the patient's position and the involvement of residents on hypoxemia during colonoscopy, but the data was probably not available to the authors.
With respect,
Author Response
Dear reviewer, we thank you for your insightful comments, and for the opportunity to revise the manuscript . We feel the manuscript is now much improved.
We present here a point-by-point response to your comments. For each comment, we copy the original text and include our responses. When changes have been made to the manuscript, we make sure to cite them here, so as to facilitate review of the changes.
Your review:
The manuscript deals with a very interesting and clinically relevant topic.
The manuscript is written in a clear and concise manner.
I think it would be interesting to investigate the impact of the patient's position and the involvement of residents on hypoxemia during colonoscopy, but the data was probably not available to the authors.
Our response:
Dear reviewer, thank you for your comment.
Indeed, these are important and interesting points. However, this data was not available. A comment regarding patients' positioning is included in the discussion section under limitations (pages 7-8, lines 238-244, text highlighted). Additionally, we have added a sentence to the limitations about the endoscopist performing the procedure.
The text in the manuscript:
For instance, detailed information on intra-procedural patient movements and posture or the precise timing of sedative administration throughout the procedure was not available. Furthermore, our database encompassed procedures performed by over 30 different endoscopists over the years. Although it was not structured to account for variations in their experience and training levels, this diversity covers a wide range of expertise, enhancing the generalizability and robustness of our results by reflecting real-world practice conditions (pages 7-8, lines 238-244, text highlighted)
Reviewer 3 Report
Comments and Suggestions for Authors
It is a retrospective study with its inherent limitations
The following are suggested to enhance the manuscript:
1. STROBE guidelines to be followed and mentioned in the methods section
2. Please explain why the hypoxemia is not related to the anaesthesia given
3. The discussion may discuss the following article:
Kim H, Hyun JN, Lee KJ, Kim HS, Park HJ. Oxygenation before Endoscopic Sedation Reduces the Hypoxic Event during Endoscopy in Elderly Patients: A Randomized Controlled Trial. J Clin Med. 2020 Oct 13;9(10):3282. doi: 10.3390/jcm9103282. PMID: 33066213; PMCID: PMC7602052.
4. The limitations of the study need to be in greater detail. Please amend
Comments on the Quality of English LanguageThe English language is largely acceptable
Author Response
Dear reviewer, we thank you for your insightful comments, and for the opportunity to revise the manuscript . We feel the manuscript is now much improved.
We present here a point-by-point response to your comments. For each comment, we copy the original text and include our responses. When changes have been made to the manuscript, we make sure to cite them here, so as to facilitate review of the changes.
Your comments:
It is a retrospective study with its inherent limitations
The following are suggested to enhance the manuscript:
1. STROBE guidelines to be followed and mentioned in the methods section
2. Please explain why the hypoxemia is not related to the anaesthesia given
3. The discussion may discuss the following article:
Kim H, Hyun JN, Lee KJ, Kim HS, Park HJ. Oxygenation before Endoscopic Sedation Reduces the Hypoxic Event during Endoscopy in Elderly Patients: A Randomized Controlled Trial. J Clin Med. 2020 Oct 13;9(10):3282. doi: 10.3390/jcm9103282. PMID: 33066213; PMCID: PMC7602052.
4. The limitations of the study need to be in greater detail. Please amend
Our response:
Dear reviewer, thank you for your comments. We checked your comments carefully and revised our manuscript according to them as far as possible. Responses are shown below.
- The sentence: This study was conducted and reported in accordance with the Strengthening the Reporting of Observational Studies in Epidemiology (STROBE) guidelines was added to the text. ( page 2 lines 81-82, text highlighted) .Guidelines were followed accordingly.
- Dear reviewer, we thank you for this comment. Accordingly, the following paragraph was added to the text at the discussion part:
Interestingly, our analysis did not find a significant direct correlation between the specific sedation protocols used (Fentanyl, Midazolam, and Propofol) and the incidence of severe hypoxemia. This observation might be explained by the fact that the administration of sedatives was personalized based on the endoscopist's discretion, allowing for variability in dosing and combination of drugs tailored to individual patient needs and responses. This personalization means that any potential effects of the sedative drugs on hypoxemia could be mitigated by the careful monitoring and adjustment by experienced clinicians. These results are consistent with a recent meta-analysis, which found that Propofol sedation poses a similar risk of cardiopulmonary adverse events compared to traditional agents used in gastrointestinal endoscopic procedures. (14) ( page 7 lines 222-231, text highlighted)
- Dear reviewer, we thank you for this comment. Accordingly, the following reference was added to the references and discussed in the discussion part:
Our study aligns with the findings of Kim et al. (13), who demonstrated that pre-oxygenation before endoscopic sedation significantly reduces the incidence of hypoxia in elderly patients.
Our data similarly underscore the multifactorial nature of hypoxemia during sedation endoscopy, reinforcing the importance of considering patient-specific factors such as age, comorbidities, and baseline respiratory function ( pages 7 lines 216-221, text highlighted)
- Dear reviewer, thank you for this comment. The limitation part was extended according to your suggestion. The added text:
Furthermore, our database encompassed procedures performed by over 30 different endoscopists over the years. Although it was not structured to account for variations in their experience and training levels, this diversity covers a wide range of expertise, enhancing the generalizability and robustness of our results by reflecting real-world practice conditions ( page 8-9, lines 240-244, text highlighted)
Reviewer 4 Report
Comments and Suggestions for Authors
Interesting study, although results are expected.
Recommendations:
• Abstract : Include specific statistical values (e.g., p-values) in the abstract to support the key findings. Formatting should be improved.
• Introduction: Expand the literature review to provide a more detailed comparison with previous studies.
• Discussion: Elaborate on the comparison with past research and further discuss the impact of unmeasured confounders.
The manuscript is well-structured and addresses a significant research question. However, it requires major revisions to enhance the clarity and depth of the literature review, discussion, and limitations sections.
Comments on the Quality of English LanguageMinor errors in language and styling.
Author Response
Dear reviewer, we thank you for your insightful comments, and for the opportunity to revise the manuscript . We feel the manuscript is now much improved.
We present here a point-by-point response to your comments. For each comment, we copy the original text and include our responses. When changes have been made to the manuscript, we make sure to cite them here, so as to facilitate review of the changes.
Your comments:
Recommendations:
- Abstract : Include specific statistical values (e.g., p-values) in the abstract to support the key findings. Formatting should be improved.
- Introduction: Expand the literature review to provide a more detailed comparison with previous studies.
- Discussion: Elaborate on the comparison with past research and further discuss the impact of unmeasured confounders.
The manuscript is well-structured and addresses a significant research question. However, it requires major revisions to enhance the clarity and depth of the literature review, discussion, and limitations sections.
Our response:
Dear reviewer, thank you for your comments.
Following your suggestions, the following changes were made:
- Abstract- the results part was revised, the new results part:
We initially collected records of 26,569 patients who underwent colonoscopy, excluding 5,045 due to incomplete data, resulting in a final cohort of 21,524 patients. Procedures under 20 minutes comprised 48.9% of the total, while those lasting 20-40 minutes made up 50.7%. Only 8.5% lasted 40-60 minutes, and 2.9% exceeded 60 minutes.
Longer procedures correlated with higher hypoxemia risk: 17.3% for <20 minutes, 24.2% for 20-40 minutes, 32.4% for 40-60 minutes, and 36.1% for ≥60 minutes. Patients aged 60-80 and ≥80 had increased hypoxemia odds (aOR 1.1, 95% CI 1.0-1.2 and aOR 1.2, 95% CI 1.0-1.4, respectively). Procedure durations of 20-40 minutes, 40-60 minutes, and over 60 minutes had aORs of 1.5 (95% CI 1.4-1.6), 2.1 (95% CI 1.9-2.4), and 2.4 (95% CI 2.0-3.0), respectively. ( page 1, text highlighted)
- Introduction- the following text was added to the introduction:
Effective strategies to mitigate this risk are critical for improving patient safety. A recent study suggested that high-flow nasal oxygen (HFNO) might offer superior outcomes in maintaining adequate oxygen levels during endoscopic procedures in high risk patients. (8) ( page 2 lines 66-69. Text highlighted)
- The discussion part was revised, and more references were added and discussed.
These paragraphs were added to the text:
These findings align with the findings of Kim et al. (13), who demonstrated that pre-oxygenation before endoscopic sedation significantly reduces the incidence of hypoxia in elderly patients.
Our data similarly underscore the multifactorial nature of hypoxemia during sedation endoscopy, reinforcing the importance of considering patient-specific factors such as age, comorbidities, and baseline respiratory function
Interestingly, our analysis did not find a significant direct correlation between the specific sedation protocols used (Fentanyl, Midazolam, and Propofol) and the incidence of severe hypoxemia. This observation might be explained by the fact that the administration of sedatives was personalized based on the endoscopist's discretion, allowing for variability in dosing and combination of drugs tailored to individual patient needs and responses. This personalization means that any potential effects of the sedative drugs on hypoxemia could be mitigated by the careful monitoring and adjustment by experienced clinicians. These results are consistent with a recent meta-analysis, which found that Propofol sedation poses a similar risk of cardiopulmonary adverse events compared to traditional agents used in gastrointestinal endoscopic procedures. (14) ( page 7, lines 219-234)
Limitations:
Furthermore, our database encompassed procedures performed by over 30 different endoscopists over the years. Although it was not structured to account for variations in their experience and training levels, this diversity covers a wide range of expertise, enhancing the generalizability and robustness of our results by reflecting real-world practice conditions. ( pages 7-8, lines 241-247, text highlighted)
Round 2
Reviewer 1 Report
Comments and Suggestions for Authors
All issues have been solved.